# Effectiveness of Promotive and Preventive Psychosocial Interventions on Improving the Mental Health of Finnish-Born and Immigrant Adolescents

**DOI:** 10.3390/ijerph19063686

**Published:** 2022-03-20

**Authors:** Kirsi Peltonen, Sanni Aalto, Mervi Vänskä, Riina Lepistö, Raija-Leena Punamäki, Emma Soye, Charles Watters, Lutine de Wal Pastoor, Ilse Derluyn, Reeta Kankaanpää

**Affiliations:** 1Department of Child Psychiatry, University of Turku, 20014 Turku, Finland; 2INVEST Research Flagship Center, University of Turku, 20014 Turku, Finland; 3Faculty of Social Sciences/Psychology, Tampere University, 33100 Tampere, Finland; sanni.aalto@tuni.fi (S.A.); mervi.vanska@tuni.fi (M.V.); riina.lepisto@tuni.fi (R.L.); raija-leena.punamaki-gitai@tuni.fi (R.-L.P.); reeta.kankaanpaa@tuni.fi (R.K.); 4School of Education and Social Work, University of Sussex, Falmer Brighton BN1 9RH, UK; e.soye@sussex.ac.uk (E.S.); c.watters@sussex.ac.uk (C.W.); 5Danish Research Centre for Migration, Ethnicity and Health, University of Copenhagen, 1353 Copenhagen, Denmark; lutine.pastoor@nkvts.no; 6Department of Social Work and Social Pedagogy, Centre for the Social Study of Migration and Refugees, Ghent University, 9000 Gent, Belgium; ilse.derluyn@ugent.be

**Keywords:** school intervention, immigrants, adolescents, mental health

## Abstract

Background: Schools are considered natural environments in which to enhance students’ social–emotional skills and mental health in general, but they can be especially important for students with refugee and immigrant backgrounds. The current study tested the effectiveness of two school-based interventions in enhancing the mental health and wellbeing of adolescents of native, refugee, and immigrant backgrounds. It further analyzed the role of age, gender, daily stressors, and discrimination in affecting the interventions’ effectiveness. Methods: A three-arm cluster RCT with parallel assignment was applied among the 16 schools. Schools were randomized to three conditions of two active interventions and a waiting-list control condition. Students (*n* = 1974) filled in an online questionnaire at baseline before the interventions, after the interventions, and at follow-up an average of 9 months after the interventions. The effectiveness criteria were internalizing and externalizing problems, resilience, and prosocial behavior. Results: Interventions were generally not effective in decreasing mental health problems and increasing psychosocial resources. The expected positive intervention effects were dependent on students’ age and gender and exposure to socioeconomic daily stressors. Conclusion: Interventions enhancing teacher awareness and peer relationships at school should be carefully tailored according to the strengths and vulnerabilities of participating students, especially their daily stress exposure, but also age and gender.

## 1. Introduction

More diverse student populations and possibilities for intercultural understanding have been issues of increasing interest in European schools in the last decade [1]. In Finland too, forced and unforced migration has brought many new families and children into the school system. Roughly, eight percent of the Finnish population has a foreign background (either one or both parents born abroad) [2]. Multicultural understanding and new skills among teachers are needed in schools to ensure successful integration, cooperation, and a sense of belonging for adolescents with diverse backgrounds.

Due to social and neurophysiological developmental changes in adolescence [3], young people are generally at heightened risk of certain mental disorders, such as depression [4,5]. Students with immigrant backgrounds are especially vulnerable, and many of them feel lonely and excluded in their peer relations when resettling in a new country [6]. Heightened levels of post-traumatic stress disorder (PTSD), anxiety, and depression are common in migrant and refugee children and adolescents [7,8,9]. These disorders are even more pronounced among refugee adolescents, as they carry multiple burdens of traumatic war and migration experiences [10].

There is thus an urgent need to provide support for native and immigrant adolescents in their everyday school and community settings by emphasizing psychological wellbeing, social affiliation, and cultural exchange [11]. Research has delineated effective school-based interventions for traumatized immigrants and refugees, especially those involving cognitive–behavioral and creative elements [11,12,13]. Yet, research is scarce on the role of teacher training and peer group interventions in enhancing adolescents’ psychological wellbeing and optimal social adjustment. The current study contributes to this research gap by examining the effectiveness of two psychosocial interventions—a teacher training and a peer group intervention—in improving mental health among native, immigrant, and refugee backgrounds.

## 2. Immigrant Adolescents’ Social Adjustment and Mental Health

Research confirms that immigrant young people (foreign-born adolescents in their new country of residence), and especially refugee adolescents, show lower wellbeing and more social and mental health problems than their native peers. Compared to native adolescents, immigrants have reported more loneliness, anxiety, and lack of close friendships, and poorer school belonging in Finland [14]. Similar results have been reported elsewhere. In an Italian study [15], immigrant adolescents showed more psychosomatic symptoms and were unhappier than their native peers. They also reported being less satisfied with their health, or life in general. Other studies have found immigrant students to be more prone both to being bullied [6,16] and to bullying others [17]. In a large-scale study including 28 European countries, the first- and second-generation migrant adolescents reported more dissatisfaction in life compared to their native peers [18]. Results on mental health symptoms have been somewhat mixed: A review concluded that migrant children in Europe showed more internalizing problems (i.e., depression, anxiety, or somatization), but not more externalizing problems (i.e., aggression, hyperactivity, or antisocial behavior) compared to native children [19]. Yet, according to a meta-analysis, both externalizing and internalizing problems of immigrant adolescents exceeded those of native adolescents [20].

Some immigrants with refugee backgrounds suffer from trauma-related problems while facing great challenges in attempting to start a new life and adjust in the host country [21]. Yet, mental health consequences of trauma differ greatly: A systematic review reported that 19–54% of refugee children and adolescents suffer from post-traumatic stress disorder (PTSD) and 3–30% from depression [22]. Demographic and immigration factors can partially explain the large differences in prevalence. Girls, older adolescents, and refugees with African (and not Asian) backgrounds tended to show more depressive and PTSD symptoms. Past traumatic events, especially separation from parents, and own injury, were associated with high PTSD. Post-migration stress such as uncertain legal status, lack of personal and institutional support, and discrimination were associated with both high PTSD and depression.

Immigrants are likely to face acculturative stress in learning new customs, rules, and expectations, and they may also experience prejudice and discrimination [23,24,25]. This may cause a risk of mental health and psychosocial problems for immigrant adolescents [26]. For example, greater exposure to acculturative stress has been found to be associated with a high level of internalizing symptoms among adolescent immigrants [25]. By contrast, integration as an acculturation strategy is associated with low levels of mental health problems and high life satisfaction and self-esteem [27]. On the other hand, multiple social and cultural factors have been found to protect immigrant adolescents’ mental health in their new home country [19,21,28]. These factors include social support, a stable and multicultural living environment, and a sense of school safety.

## 3. Effectiveness of Psychosocial School Interventions

Several meta-analyses and systematic reviews have been conducted on interventions focusing on improving students’ mental health and wellbeing in high-income countries. Some have focused on native-born adolescents [29,30], and others on immigrant adolescents and refugees [11,13,31,32]. The evidence concerning the reduction in PTSD, functional impairment, and peer problems among adolescents participating in professionally conducted, clinical treatments is promising. However, results are still ambiguous concerning universal promotive and preventive whole-school or classroom-based interventions. Whole-school interventions have struggled to show effectiveness because of challenges in engaging all school staff, a lack of clear-cut intervention aims, and difficulties in data collection [29,33].

Teacher training and awareness raising are common whole-school intervention approaches. A review summarized the effects of teacher training programs that aimed to increase teachers’ competence in addressing students’ mental health needs [34]. Most of the reviewed studies were successful in increasing knowledge among teachers and some also found improvement in teachers’ ability to intervene and refer students to appropriate services. A review by Stratford et al. [35] reported that successful teacher training elements included improved counseling and communication skills, increased knowledge and awareness about traumatic experiences in learning, and more active parent engagement (the criterion of success was effective support for trauma-exposed students).

It has been suggested that teachers’ motivation and cultural competence play an important role in supporting the mental health of refugee and migrant students [36,37]. Teacher self-efficacy, i.e., the teachers’ own judgement of their capability in inducing desirable student outcomes, has also been found to improve student academic performance [38] and motivation through instructional support [39]. A systematic review analyzed the effectiveness of teacher training in increasing pre -and in-service teachers’ cultural competence, indicated by attitudes, knowledge, and practices of teaching multicultural classes [40]. Results showed that the most effective teacher trainings were characterized by group-guided critical reflection, enactment, and embedded training in schoolwork. Another review did not report empirical assessments of the effectiveness of whole-school approaches delivered by non-clinical staff [35]. However, it emphasized the importance of teachers’ cultural competence and community engagement in addressing the needs of trauma-exposed students in schools. All reviewed studies focused on teachers’ professional development and cultural competence, but they did not study students’ wellbeing or academic performance as effectiveness outcomes. Finally, the review by Ohrt et al. [34] concluded that most of the teacher training programs were successful in increasing knowledge about mental health among teachers; some studies in the review also found improvement in teachers’ ability to intervene and refer students to appropriate services.

We could not find studies on the effectiveness of classroom-based interventions in enhancing peer relations or social support among refugee or migrant youth, using mental health or wellbeing as effectiveness criteria. According to a systematic review, peer-facilitated community-based intervention trials in low- and middle-income countries have shown mixed results [41]. Yet, there is some promising evidence to support the effectiveness of working with peer facilitators to improve adolescent mental health. Fifteen out of 20 studies in the review were conducted at schools or a college, and five in community settings, which may indicate the importance of supportive peer work, especially in a classroom context.

In sum, there is lack of research studying the effects of school-based interventions, focusing either on teacher competence or peer relations, on adolescents’ wellbeing. This is especially true concerning the samples with native and immigrant students. In this study, we aim to find out whether interventions that aim to improve teachers’ competence and self-efficacy or peer relations among students in diverse classrooms are effective in supporting students’ mental health.

## 4. Study Aims

The first aim of our study was to examine whether two school-based psychosocial interventions are effective in improving the mental health and psychosocial wellbeing of adolescents. The interventions were: (a) In-service teacher training (INSETT) and (b) peer integration and enhancement resource (PIER). Their effectiveness was evaluated separately among Finnish-born students and immigrant students (with immigrant or refugee backgrounds). The criteria for the interventions’ effectiveness were reduced mental health problems, i.e., internalizing and externalizing symptoms, and increased resilience and prosocial behavior (i.e., factors promoting psychosocial wellbeing). We hypothesized that internalizing and externalizing problems would decrease statistically significantly among adolescents participating in INSETT and PIER interventions. Likewise, we hypothesized that prosocial behavior and resilience would increase among adolescents participating in INSETT and PIER interventions during the measurement period. Furthermore, we expected that in the control group the levels of internalizing and externalizing problems, prosocial behavior, and resilience would remain unchanged. The timeline was from baseline (T1), through post-intervention at six months (T2), to follow-up at 12 months (T3). The second aim was to explore whether the interventions had different impacts on immigrant vs. Finnish-born students, and whether gender, age, socioeconomic daily stressors, or discrimination experiences were associated with intervention effectiveness (i.e., affected the decrease in problems and increase in resources from T1 through T2 to T3). 

## 5. Materials and Methods

### 5.1. Study Setting and Data Collection

The study was a three-arm clustered parallel assignment, quasi-randomized, controlled trial (RCT) comparing INSETT and PIER intervention groups to the waitlist control group. The recruitment of intervention schools was based on three principles. First, schools participated voluntarily—in other words, the schools (i.e., their headmasters and teachers) expressed a self-defined need for an intervention to improve the wellbeing of refugee and immigrant students, to promote more positive interethnic peer relations, or to increase teachers’ multicultural awareness. Second, schools had a considerable share of students from refugee and immigrant backgrounds, estimated as 30–50 percent in each class. Third, the schools accepted the possibility of serving as a waitlist control school until the next school year when they would be provided the INSETT intervention in an electronic form.

The recruitment commenced by sending emails to Finnish secondary schools through mailing lists and by advertising the study in a teachers’ vocational magazine. The first wave of recruitment was complemented with around 100 phone calls to headmasters of schools residing in the metropolitan or other immigrant-dense areas in larger Finnish cities. In the second wave of recruitment, schools that were located near asylum centers were approached by phone calls and e-mail communications. The only exclusion criterion for schools was having other similar psychosocial interventions running at the same time. The schools that enrolled in the study were located across Finland, and they represent mainly refugee- and immigrant-dense urban areas. One school was in the south (the metropolitan area), two located in the west, three in the east, three in the north, and seven in the center of Finland.

The 16 schools were randomized in a parallel-group condition with an allocation ratio of 1:1:1. The voluntary schools were first grouped based on whether they offered an introductory class for newly arrived students or not. The schools in the two groups were then randomly allocated to either intervention or control conditions using a random number generator by StatsDirect [42]. The allocation aimed at equal numbers of schools in each condition. Due to the low participation rate, recruitment of schools continued, which resulted in three additional schools, all with introductory classes. The three additional schools were thus randomized either into the INSETT or control group. Following the randomization, eight schools were allocated into the INSETT intervention, three schools were allocated to PIER, and five schools served as waiting-list controls.

Figure 1 presents the data collection time frame and numbers. Baseline data were collected mainly before the implementation of the INSETT and PIER interventions between August 2019 and January 2020. The post-intervention (T2) data collection took place from February (9th grade) to May 2020 and the follow-up (T3) data from May (9th grade) to October 2020. Adolescents responded online to the questionnaires in their mother tongue, guided at baseline by researchers and teachers in classrooms. Due to the COVID-19 school closures, the T2 and T3 data collections were guided online. To secure anonymity, a secure online survey tool (Lime Survey GmbH) was used, and all identifiable information was stored separately from the research data. To avoid attrition in the post-intervention (T2) and follow-up (T3) surveys, students received a personal invitation and two reminder letters via e-mail. For students, there were no exclusion criteria and all signing the informed consent could participate. Online INSETT teacher training material was provided to waiting-list control schools in January 2021.

### 5.2. Sample Characteristics

In recruitment, school authorities estimated the number of students and the proportion of foreign-born students in all participating classes. Schools do not have information on the migration backgrounds of students. Hence, the categorization is based on self-reported reason for migration and country of origin. We invited all children in schools to participate. The actual number of students that were invited to participate in the study was 3062, with 19% being students with immigrant backgrounds. Yet, 1974 students, of whom 223 were foreign-born and 1770 Finnish-born secondary school students (aged 12–17 years), answered the baseline (T1) questionnaire. While over two-thirds (71%) of the invited Finnish-born students participated, refusal among immigrant students was considerable; only four out of ten (39%) participated. It is noteworthy that a total of 1441 students were invited from INSETT schools, but only 70% of them filled in the baseline questionnaires and could therefore be analyzed in the effectiveness study. Additionally, a total of 374 students were invited from PIER schools, but only 81% of them filled in the baseline questionnaires and could therefore be analyzed in the effectiveness study.

The attrition from baseline to T2 was approximately 31% among Finnish-born and 52% among foreign-born students in INSETT, 20% among Finnish-born and 41% among foreign-born students in PIER, and 43% among Finnish-born and 39% among foreign-born students in the control group. A quarter (27%) of teachers who participated in the INSETT seminars, and half (50%) of teachers who participated in the PIER training, also participated in the questionnaires.

### 5.3. The Interventions

In-service teacher training (INSETT) was developed by Lutine de Wal Pastoor at the Norwegian Centre for Violence and Traumatic Stress Studies, and is designed for lower and upper secondary school teachers and counselors in introductory, preparatory, and ordinary classes [43]. INSETT aims to strengthen teachers’ competence and self-efficacy in three areas. First, promoting refugee/migrant students’ mental health and psychosocial wellbeing. Second, encouraging positive interethnic relationships and strengthening school belonging. Third, fostering supportive interrelationships with parents, caregivers, and/or guardians to promote school involvement [43]. The INSETT intervention consists of three interrelated course modules. It combines lectures, group work, and exchange of participants’ experiences, views, and reflections in two whole-day seminars with individually completed online training “sandwiched” between the two seminars (an introductory seminar and a follow-up seminar, respectively). The INSETT intervention manual working paper [43] introduces and elaborates on the aims and content of the intervention, and describes the content of the seminars. The online teacher training course used in INSETT, *Providing support to refugee youth*, developed by the Augeo Foundation in the Netherlands (Augeo Academy) was translated into Finnish. The INSETT manual working paper [43] includes a full manuscript and a PowerPoint presentation for the lecture “Young refugees’ psychosocial challenges upon resettlement: the need for a refugee-competent school”. For implementation in the RefugeesWellSchool project, additional PowerPoint presentations were developed on two topics suggested in the INSETT manual working paper but for which the manual does not include manuscripts, i.e., cultural competence and trauma impact and stabilization.

Peer integration and enhancement resource (PIER) was designed by Charles Watters and Emma Soye at the University of Sussex [44]. PIER aims at supporting safe and positive peer interactions and social relationships in multiethnic schools. This is expected to occur through group exercises that focus on strengthening belonging, empathy, and role-taking; learning from each other; and giving and receiving more social support. The manualized intervention consists of eight sessions ranging from 45 to 90 min. Sessions include structured welcoming and ending rituals and multimodal group activities such as cartoon drawing, role-play, movies, and drama. Sessions also include processes of reflection on various identities, migration, and racism. The group facilitators were secondary school teachers, and they delivered the PIER intervention in classes including both Finnish-born and refugee and immigrant students. The PIER facilitators participated in a two-day training. The first day consisted of practicing each intervention session together and familiarizing teachers with the resource material through lectures and discussion. The second day involved sharing experiences and valuable ideas to improve the intervention as well as participation in focus group interviews.

### 5.4. Intervention Dosage

In two schools, all teachers participated in both INSETT seminars, which were organized in the school facilities. In the remaining six schools, only interested teachers participated in the INSETT seminars, which were organized in two towns in Finland. In the PIER intervention schools, the teachers participated in the PIER training organized collectively for all schools across Finland. The 16 schools that participated the study had in total 452 teachers, of which 252 were in the INSETT schools, and 81 were in the PIER schools. Altogether, 98 teachers participated in the first INSETT seminar, and 82 in the second. Only 12 teachers completed the Augeo online course. Regarding the PIER intervention, 16 teachers participated in the training and nine of them participated in the feedback session. Teachers facilitating the PIER intervention were asked to fill in structured intervention protocols and to keep a log of the tools and methods they had carried out in the class. The protocols were used to monitor the implementation process. The PIER intervention also included a logbook to measure implementation fidelity. Unfortunately, only one-third of the teachers returned the logbooks. Six teachers participated in the PIER feedback session.

### 5.5. Ethical Considerations

The study was registered with ISRCTN (ISRCTN64245549) and the study protocol was published before the data analysis [45]. The final study reported in this article differs from the research protocol in two ways: First, due to a very small sample size of third intervention teaching recovery techniques (TRTs), we are not able to provide a quantitative analysis of the intervention (including the outcome post-traumatic stress symptoms. Second, because of the low response rate of teachers in the INSETT intervention and immigrant students in the PIER intervention we are not able to analyze the mediators of these interventions. The Ethics Committee for Human Sciences of Tampere University reviewed and approved the study setting, recruitment, informed consents, and measurement (Code #9/3/2019) in addition to the Horizon 2020 Ethical Review. The Horizon 2020 research team prepared informed consent documents in 18 languages, which guaranteed that all students and parents were informed about the project and their participation rights. Informed consent was obtained from all study participants and the translations offered the participants the possibility of signing the agreement in their mother tongue. The informed consent signed at T1 also included an agreement for participation at T2 and T3. The project complies with the Code of Ethics of the World Medical Association (1964 Declaration of Helsinki and its later amendments). In addition to the two Ethics Committee approvals, we requested and obtained research permission for all participating schools from the municipality school administration and the headmasters.

### 5.6. Measures

Mental health problems were measured with a self-report version of the Strengths and Difficulties Questionnaire (SDQ) for 11–17-year-olds [46]. The SDQ has five subscales, each with five items. Problems scales are Emotional problems (e.g., “I am often unhappy, sad or tearful”), Conduct problems (e.g., “I get very angry and often lose my temper”), Inattention–hyperactivity (e.g., “I am constantly fidgeting or squirming”), and Peer problems (e.g., “I am usually on my own”). The fifth scale is called Prosocial behavior, reflecting strengths (e.g., “I usually share with others (food, games, pens, etc.)). Adolescents responded using a 3-point scale on the extent to which they found the claim to be true (“not true”, “somewhat” or “certainly true”).

For the current study, we calculated sum scores for internalizing Emotional problems and Peer problems and externalizing Conduct problems and Inattention–hyperactivity. Their Cronbach’s alpha coefficients in the baseline were α = 0.77 and α = 0.77 for Finnish-born students, and α = 0.71 and α = 0.67 for immigrant students, respectively. The alphas for the post-intervention (T2) were α = 0.76 and α = 0.76 for Finnish-born, and α = 0.77 and α = 0.70 for immigrant students. For follow-up (T3), they were α = 0.79 and α = 0.77 for Finnish-born, and α = 0.78 and α = 0.78 for immigrant students. The alpha for Prosocial behavior was at baseline α = 0.66 for Finnish-born students, and α = 0.68 for immigrant students. In T2, it was α = 0.68 for Finnish-born, and α = 0.78 for immigrant students. Finally, in T3, it was α = 0.66 for Finnish-born students, and α = 0.72 for immigrant students.

Research shows satisfying psychometric properties of the SDQ total score among children and adolescents in different western countries, including Finland [47]. Among immigrants, some researchers have recommended using the total SDQ score. Others have recommended using the SDQ with caution among children and adolescents of refugee background [48,49].

Resilience was measured with the 12-item reduced Child and Youth Resilience Measure (CYRM-12) [50]. Being similar to the original 28-item CYRM, it measures individual, relational, communal, and cultural resources that may bolster resilience among youth between 9 and 23 years of age. Adolescents evaluated on a 5-point scale how well the items described them (1 = “not at all”, 2 = “a little”, 3 = “somewhat”, 4 = “quite a bit”, and 5 = “a lot”). A total score was a sum score of the items, with a Cronbach’s alpha coefficient of α = 0.85 for Finnish-born and α = 0.84 for immigrant students, at the baseline. In the post-intervention, it was α = 0.85 for Finnish-born and α = 0.86 for immigrant students. Finally, in the follow-up, α = 0.86 for Finnish-born students and α = 0.77 for immigrant students. Research shows some content validity of the CYRM-12 to merit its use as a screener for adolescents’ resilience processes [50].

Discrimination was measured with five dichotomous items asking whether the respondent has experienced discrimination due to specific attributes (e.g., the color of their skin, or being a refugee or immigrant). If one or more items included the response “yes”, the respondent had a value “1” on the variable, otherwise “0”.

Daily stressors were measured with the 6-item Daily Stressors Scale for Young Refugees (DSSYR; unpublished) [51]. The questions start with a common sentence “How often does this happen?” and relate to having enough food, clothing, money, a place to live, enough medical care, and safety. The adolescents assessed how often they experienced these conditions (1 = “never”, 2 = “sometimes”, 3 = “often”, 4 = “always”, and 5 = “I do not know/I do not want to answer”). The options from 1 to 4 were calculated into a sum score and reversed so that a higher value refers to more stressors. The validity or reliability of DSSYR has not yet been evaluated.

For demographic information and immigrant background, adolescents reported their gender (boy, girl, other) and age (open response). The immigrant background was measured with the following questions: Whether the student was born in Finland or not (yes/no), the country of birth (open response), and reasons for immigration (fleeing war, danger, or persecution, parents’ permanent/temporary work, family reunification). For the current study, adolescents were categorized as Finnish-born or immigrants. The students were classified in these groups based on their reports on their home country and reasons for migration. In instances where the participant did not report the reason for migration (*n* = 91 at T1, 59 at T2, and 52 at T3), information about their home country and statistics from Finnish Immigration Services between 2015 and 2021 on residence permits (international protection) and safety classification on countries was used for the allocation.

### 5.7. Analysis

Data were collected in Lime Survey in a long form, i.e., each participant’s response was unique. Participants received an identity number which then was used to combine responses from different assessment points. Identity numbers in a data file were checked manually for errors so that the correct responses from the same respondent were brought together. Each timepoint was then merged one by one so that we ended up with wide format data of three timepoints.

Due to a small number of participating schools, analyses were conducted between students. The results were analyzed separately in the INSETT and PIER intervention groups as well as in Finnish-born and immigrant students. The immigrant students include both refugees and other immigrant students. However, the sample sizes in these separate groups were so small that the results ought to be interpreted with caution. The results from the PIER intervention are reported as one group of students because the sample size of foreign-born students was too small for parametric analysis.

Due to the substantial range in the response times in each timepoint, we used the growth curve model (GCM) with individually varying timepoints of observation (so-called unstructured GCM) which is available in Mplus 8.0 software [52]. Due to varying timepoints of observation, the unstructured GCM has produced less biased results in several simulation studies [53,54,55]. The unstructured GCM is a multilevel model and also allows taking clustering into account. The response times were coded as weeks from the start of the data collection.

In the traditional GCM, the observed outcome variables in each timepoint have fixed loadings on two latent variables, i.e., intercept and slope. On the intercept, all the loadings are fixed on 1. On the slope variable, baseline response is fixed on 0, and timepoints 2 and 3 are fixed on values 1 and 2, assuming linear change. In the unstructured GCM, however, the response time impacts all loadings. For instance, the later the response is given, the lower/higher the level is assumed to be. It follows that the variance of the outcome variables changes as a function of response time values [56]. This means that there is no constant model implied matrix Σ to test the model fit. Therefore, the chi-square statistic cannot be calculated, and thus the traditional fit statistics are not available. Nevertheless, the unstructured GCM could be compared with the traditional GCM using Akaike information criterion (AIC) and Bayesian information criterion (BIC) indices. In most of the models, and especially in the migrant group, the unstructured model showed a better fit than the traditional GCM. The latent growth parameters were estimated using a robust maximum likelihood estimator (MLR), which allows the use of missing data without any separate imputations and non-normal continuous indicators to be reliably analyzed.

## 6. Results

### 6.1. Descriptive Statistics

Table 1 shows the numbers of participants at baseline. Altogether, 995 students participated in research in the INSETT intervention schools. Of the students, 89% were Finnish-born and of them 49% were boys. Of 106 immigrant student participants in the INSETT schools, 55% were boys. The mean age of the INSETT participants was 13.9 years among Finnish-born students and 14.5 years among immigrant students. Out of 731 participants in the INSETT control schools, 85% were Finnish-born and of them 48% were boys. Out of 59 immigrant participants in the INSETT control schools, 49% were boys. The mean age of the INSETT control participants was 13.9 years among Finnish-born students, and 14.6 years among immigrant students.

Out of 108 participants in the *PIER intervention,* 86% were born in Finland, and of those, 51% were boys. Out of 15 immigrant students participating in the PIER intervention, nine were boys. The mean age of the PIER participants was 13.6 among Finnish-born students and 14.3 among immigrant students. Out of 187 participants in the *PIER control group,* 87% were born in Finland, and of those, 42% were boys. Out of 24 immigrant participants in the PIER control group, seven were boys. The mean age of the PIER control participants was 13.9 among Finnish-born students and 14.2 among immigrant students. The estimates for all independent variables are shown in Table 2. The estimates for all dependent variables at T1, T2, and T3 are shown in Table 3.

### 6.2. Intervention Effects

Table 4 shows the results regarding the INSETT intervention on mental health and psychosocial wellbeing for Finnish-born students and Table 5 shows immigrant students. Contrary to our hypothesis, the non-significant direct intervention effects indicate that externalizing or internalizing problems did not decrease, nor did resilience and prosocial behavior increase among Finnish-born or immigrant students in the INSETT intervention group.

Concerning our second, explorative research aim (whether gender, age, socioeconomic daily stressors, or discrimination experiences were associated with intervention effectiveness), the results show significant moderating interaction effects among both Finnish-born and immigrant student groups. Of background variables, the students’ age, daily stressors, and discrimination experiences were associated with the INSETT intervention’s effectiveness (indicated by decrease in externalizing and internalizing problems and increase in prosocial behavior and resilience). However, the moderating effects were different among Finnish-born and immigrant students. Among *older* Finnish-born students, the internalizing problems decreased significantly more in the intervention than in the control group (β = −0.014, *p* < 0.000 for age * intervention interaction). Among Finnish-born students exposed to a high level of *daily stressors,* internalizing problems increased (β = 0.016, *p* < 0.05 for daily stressors * intervention interaction), and among immigrant students exposed to high levels of daily stressors, externalizing problems increased (β = 0.048, *p <* 0.000 for daily stressors * intervention interaction). Meanwhile, the same negative change was not found in the control group. In the INSETT intervention group, prosocial behavior increased among immigrant students who had high levels of discrimination experiences (β = 0.027, *p* < 0.001 for discrimination experiences * intervention interaction), while prosocial behavior decreased (β = −0.016, *p <* 0.05 for discrimination experiences * intervention interaction) among Finnish-born students with high levels of discrimination experiences. Yet, among immigrant students with high levels of discrimination experiences, externalizing symptoms increased (β = 0.015, *p* < 0.05 for discrimination experiences * intervention interaction) in the INSETT intervention. The intervention effectiveness was not dependent on gender.

### 6.3. PIER

As explained in the analysis section, the results from the PIER intervention are combined across Finnish-born and immigrant students, due to a low number of participants. The results in Table 6 reveal that, contrary to our hypothesis, the non-significant direct effects indicate that internalizing and externalizing problems did not decrease, nor did resilience and prosocial behavior increase in the PIER intervention group. As the non-significant immigrant status * intervention interaction effects indicate, the PIER intervention’s effectiveness did not differ between Finnish-born and immigrant students.

Of the moderating background variables, only adolescents’ age and gender, but not discrimination experiences or daily stressors, were associated with the PIER intervention’s effectiveness. Gender emerged as important for intervention effects on externalizing symptoms and resilience, showing that the PIER intervention was not effective among girls. On the contrary, among girls, externalizing symptoms increased (β = 0.010, *p* < 0.01) and resilience decreased (β = −0.084, *p* < 0.05) in the intervention group, but not in the control group (see Table 6, concerning gender * intervention interaction effects). Concerning age, internalizing (β = −0.018, *p* < 0.000) and externalizing (β = −0.015, *p* < 0.01) symptoms decreased and resilience increased (β = 0.025, *p* < 0.000) significantly more in the intervention than in the control among older students.

## 7. Discussion

We sought to study whether two school-based psychosocial interventions are effective in improving the mental health and psychosocial wellbeing of secondary school students, and whether the interventions had a different impact on immigrant vs. Finnish-born students. We also explored the role of students’ background and socioeconomic stressors in the interventions’ effectiveness. The results showed, unexpectedly, that neither the INSETT nor the PIER interventions were generally effective in decreasing mental health problems or increasing psychosocial resources. The expected positive findings were dependent on students’ gender and age. The results also suggested that students’ daily stressors and discrimination experiences can have a decisive influence on the interventions’ effectiveness. Unfortunately, due to teachers’ low participation rate in INSETT and students’ low participation in the PIER interventions, we were not able to study the mediation effects as planned. This means that we cannot analyze whether the observed effects follow the hypothesized paths from interventions to the increased cultural competence of teachers and interethnic friendships of students’ and further to the better mental health and wellbeing of students. Our results should be considered as rough indicators of the effectiveness of school-based interventions among Finnish-born and immigrant secondary school students.

The positive effects of the interventions include the possible protective effect of the INSETT intervention against internalizing problems among older Finnish-born students and increasing prosocial behavior among immigrant students who are discriminated against. It is possible that the enhancement in teachers’ competence and self-efficacy encouraged them to pay more attention to their students’ psychosocial- and mental health-related challenges. Interestingly, the effect of age was evident only among Finnish-born students, and the INSETT intervention had positive effects in especially high-risk groups of immigrant students, i.e., those suffering from discrimination. The findings are important since they suggest that intervention themes concerning multicultural peer relationships and trauma-informed pedagogy are relevant not only for the wellbeing of immigrant and refugee students, but for the wellbeing of all students. Although the main aim of the INSETT intervention was to promote psychosocial support and social cohesion for immigrant and refugee adolescents through teachers’ education, there may have been a more general change in teachers’ diversity competence in the school environment, which had an effect on the wellbeing of older students and those with discrimination risks.

Some positive effects were also found concerning the PIER intervention, yet they depended on students’ gender and age. Results show that internalizing and externalizing problems decreased, and resilience increased in the intervention among older students participating in the PIER intervention, but not in the control group. In other words, older students and boys particularly benefited from the PIER intervention, compared to girls and younger students. Adolescence is a time of intensive changes in peer relationships, worldview, and identity formation (4), and it is possible that the older students found it easier to absorb the intervention material relating to safe and positive peer interactions. Improved mental health in multiethnic schools may reflect the core aims of the PIER intervention, which include a sense of belonging, empathy, curiosity to different experiences, and providing social support to others. Due to a low number of participants, we were unfortunately not able to examine the effects separately for Finnish-born and migrant students.

Our results further showed that neither the INSETT nor the PIER interventions were effective in improving mental health among adolescents who experienced severe daily stressors, such as not having enough food, clothing, or safety. On the contrary, in the INSETT intervention, externalizing symptoms increased among students with severe daily stress more than in the control group. Similarly, prosocial behavior decreased in the INSETT intervention among Finnish-born students who had discrimination experiences. It is possible that teachers’ multicultural competence or self-efficacy did not improve in a way that would have helped highly stressed and discriminated against adolescents to benefit from their new methods in the classroom, but rather vice versa.

Like other intervention studies among war-affected adolescents [57], our results show gender differences. The PIER intervention seems to be somewhat ineffective among girls, as their externalizing problems increased and resilience decreased in the intervention but not in the control group. We could not find an earlier explanation for gender differences from the literature, but a very preliminary interpretation could be that the “free” curriculum activities and the concomitant possibility of losing behavioral boundaries as part of the PIER intervention may not suit girls, causing externalizing symptoms in particular to increase.

The limits of our study concur with the critics of earlier findings of psychosocial school-based interventions. Both Paulus et al. [33] and Goldberg et al. [29] argued that whole-school interventions struggle to show effectiveness partly because of the challenges of intervention design and successful data collection. In our study too, the initial research aims of searching the mediating mechanisms and analyzing all three levels of interventions were not fulfilled. Cipriano et al. [58] suggested that educational interventions targeting classroom processes should be enacted at the school level and assessed by both teachers and students to capture the breadth of perceptions, and to optimize outcomes among adolescents. Furthermore, only 39% of the immigrant students we contacted were willing to participate in the study, which makes it difficult to generalize the findings outside the population in this trial. This sets clear standards for future research.

In our study, the onset of the COVID-19 pandemic during the post-intervention and follow-up assessments posed additional challenges for these endeavors. Teachers’ response rate to the baseline survey was dramatically low, and we could not analyze their data. The absence of their data is unfortunate but not surprising according to previous studies [59]. In future trials, support for intervention feasibility should be enhanced among teachers. We could not obtain satisfactory reports from all teachers of how the interventions were implemented in the classroom. Due to the COVID-19 pandemic, this is quite understandable since the teachers struggled with all kinds of extra demands in their everyday schoolwork. Additionally, the percentage of immigrant students who participated in the study should have been more in order to make strong implications on the differences between the students with different backgrounds. Having more resources to ensure the participation of immigrant students should be taken into consideration in future research. Luckily we were able to report the qualitative results of TRT, and teachers’ and parents’ insight of the interventions for the whole RefugeeWellSchool program in the effectiveness report [60]. We can only speculate on the other effects COVID-19 might have had on our results. Finally, study designs can be further refined in trials like this. In the current study, blinding of participants, intervention providers, outcome assessors, or data analysts was not possible due to the explicit nature of the interventions and the low number of research staff.

## 8. Conclusions

The positive effects of interventions in this study were dependent on students’ characteristics such as gender, age, exposure to daily stressors, and their discrimination experiences. This means that tailored help is needed in order to support adolescents’ wellbeing at schools. Future studies should continue to search for critical factors that affect the suitability of the interventions for adolescents with different backgrounds. A multilayered intervention study with voluntary participation can be difficult to conduct with the requirements of a randomized controlled trial. Our study shows that to some extent the experimental setting is still achievable. We did not find evidence for teacher- or peer-related programs increasing students’ wellbeing at school. It is, however, possible that in some circumstances and among some student groups these interventions might be effective, indicated by gender- and age-specific effects, as well as the role of students’ daily stressors and discrimination experiences in the interventions’ effectiveness.

## Figures and Tables

**Figure 1 ijerph-19-03686-f001:**
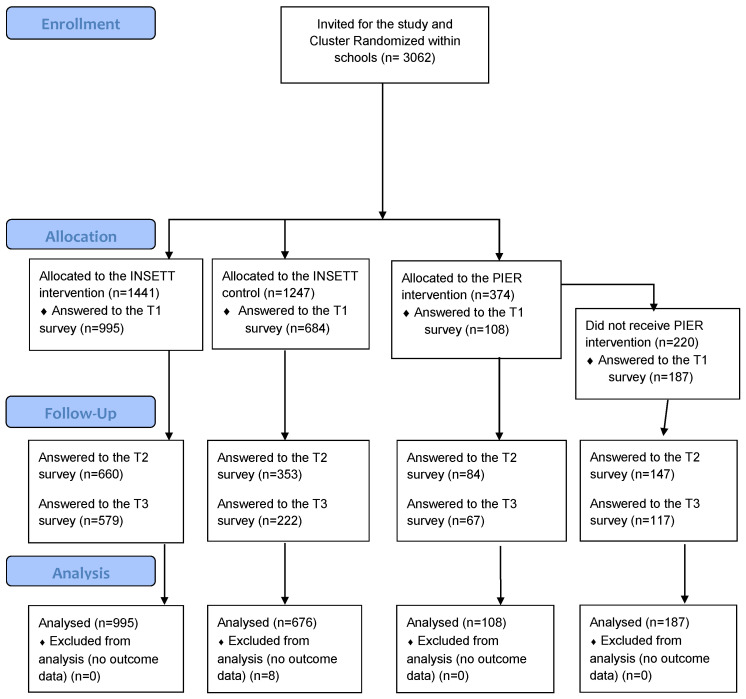
Accepted CONSORT 2010 Flow Diagram.

**Table 1 ijerph-19-03686-t001:** Numbers of participants at baseline.

Groups	*n*
All students in INSETT intervention schools	2940
Students with immigrant background at INSETT schools	386
All students invited to the study at INSETT schools	1441
Students with immigrant background invited to the study in INSETT schools	223
Native students responding to T1 survey at INSETT schools	889
Immigrant students responding to T1 survey at INSETT schools	114
All students in INSETT control schools	1546
Students with immigrant background at control schools	151
Students invited to the study in control schools	1247
Students with immigrant background invited to participate in the study in control schools	214
Students responding to T1 survey in control schools	624
Immigrant students responding to T1 survey in control schools	62
All students in PIER intervention and control classes	922 ^a^
Students with immigrant background in PIER intervention and control classes	163
All students invited to the study in PIER classes	374
Students with immigrant background invited to the study in PIER classes	134
Native students responding to T1 survey in PIER classes	96
Immigrant students responding to T1 survey in PIER classes	24
All students in PIER control classes	220
All students invited to the study in PIER control classes	187
Native students responding to T1 survey in PIER control classes	96
Immigrant students responding to T1 survey in PIER control classes	24

^a^ Participants in the PIER intervention were not randomized into control classes, and thus the T1 number is a combination of both intervention and control class students.

**Table 2 ijerph-19-03686-t002:** Estimates for independent variables.

Interventions	N	Min	Max	Finnish-Born	Immigrants
INSETT intervention	995			889	106
Boys	496			438	58
Girls	477			430	47
Experienced discrimination (%)	110			65 (7.5%)	45 (44.1%)
Age M (SD)	920	13	17	13.87 (0.779)	14.525 (1.045)
Daily stressors M (SD)	985	1	4	1.217 (0.393)	1.394 (0.623)
T1 response week (M SD)	995	0	14	3.90 (4.953)	3.02 (4.349)
T2 response week (M SD)	660	25	41	29.13 (4.898)	30.31 (4.641)
T3 response week (M SD)	579	38	57	51.24 (5.859)	47.63 (7.958)
INSETT Control group	731			625	59
Boys	331			302	29
Girls	336			308	28
Experienced discrimination (%)	86			64 (10.7%)	22 (40.7%)
Age M (SD)	612	13	17	13.885 (0.825)	14.596 (1.245)
Daily stressors M (SD)	665	1	4	1.258 (0.435)	1.330 (0.406)
T1 response week (M SD)	684	1	14	2.78 (1.279)	2.83 (1.652)
T2 response week (M SD)	353	25	42	29.93 (2.827)	27.74 (3.048)
T3 response week (M SD)	222	39	64	53.76 (8.634)	51.70 (9.033)
PIER intervention	108			93	15
Boys	56			47	9
Girls	47			43	4
Experienced discrimination (%)	15			10 (11.5%)	5 (41.7%)
Age M (SD)	93	13	17	13.551 (0.714)	14.267 (0.594)
Daily stressors M (SD)	107	1	4	1.268 (0.469)	1.213 (0.401)
T1 response week (M SD)	108	1	3	1.11 (0.454)	1.27 (0.704)
T2 response week (M SD)	84	24	28	26.70 (1.885)	25.60 (2.066)
T3 response week (M SD)	67	54	57	54.32 (0.567)	54.29 (0.488)
PIER Control group	187			163	24
Boys	75			68	7
Girls	108			92	16
Experienced discrimination (%)	26			14 (9.5%)	12 (70.6%)
Age M (SD)	169	13	17	13.892 (0.834)	14.191 (1.123)
Daily stressors M (SD)	181	3	4	1.177 (0.256)	1.139 (0.193)
T1 response week (M SD)	187	1	4	2.06 (0.788)	2.25 (0.847)
T2 response week (M SD)	147	24	29	28.21 (1.669)	28.06 (2.016)
T3 response week (M SD)	117	39	57	49.08 (7.909)	50.00 (8.261)

**Table 3 ijerph-19-03686-t003:** Estimates for dependent variables.

			Finnish-Born									Immigrants								
			T1			T2			T3			T1			T2			T3		
	Min	Max	*n*	Mean	SD	*n*	Mean	SD	*n*	Mean	SD	*n*	Mean	SD	*n*	Mean	SD	*n*	Mean	SD
INSETT																				
Internalizing	0	20	883	4.99	3.42	609	5.09	3.33	501	5.02	3.53	104	5.53	3.37	46	5.35	3.81	62	5.87	3.74
Externalizing	0	20	883	4.90	3.26	609	4.75	3.16	501	4.53	3.14	104	4.83	2.76	46	5.22	3.8	62	5.21	3.73
Prosocial	0	10	883	7.23	1.88	609	7.39	1.93	501	7.42	1.94	104	7.63	1.95	46	7.09	2.11	62	7.26	2.35
Resilience	12	60	854	46.83	7.83	586	47.18	7.53	478	47.48	7.79	87	45.22	9.00	46	44.10	9.52	59	46.32	7.20
INSETT Control																				
Internalizing	0	20	603	4.89	3.59	304	5.05	3.52	191	5.69	3.77	56	6.20	3.68	32	6.69	3.54	18	6.78	3.64
Externalizing	0	20	603	4.67	3.40	304	4.41	3.30	191	4.51	3.30	56	5.57	3.12	32	4.72	3.11	18	4.72	3.12
Prosocial	0	10	603	7.42	1.85	304	7.60	1.68	191	7.49	1.64	56	7.48	1.98	32	7.94	1.79	18	7.56	1.92
Resilience	12	60	579	47.35	6.91	287	47.65	6.44	180	48.10	6.90	53	46.40	6.46	28	46.29	6.63	17	47.71	5.61
PIER																				
Internalizing	0	20	90	4.13	3.16	73	4.37	2.97	58	4.12	2.99	15	3.47	2.95	10	6.20	5.27	7	4.86	5.96
Externalizing	0	20	90	4.74	3.43	73	4.86	3.40	58	4.60	3.42	15	4.13	3.64	10	6.30	2.95	7	3.86	3.98
Prosocial	0	10	90	7.40	1.83	73	7.26	1.81	58	7.29	1.76	15	7.53	1.96	10	6.20	2.78	7	8.71	1.11
Resilience	12	60	87	47.29	6.23	68	47.72	6.46	57	48.05	7.82	11	52.27	3.90	10	40.70	12.73	7	53.14	4.56
PIER Control																				
Internalizing	0	20	157	4.96	3.62	131	5.85	3.83	107	5.03	3.64	17	6.47	3.36	15	7.27	4.33	9	6.67	3.87
Externalizing	0	20	157	4.85	3.14	131	5.09	2.97	107	4.69	3.39	18	6.56	4.00	15	5.80	3.61	9	5.78	5.04
Prosocial	0	10	157	7.49	1.68	131	7.31	1.66	107	7.68	1.74	18	7.00	2.09	15	6.53	3.27	9	7.56	2.30
Resilience	12	60	148	47.84	6.73	123	46.74	6.98	104	48.11	6.45	14	43.43	9.39	12	45.92	7.25	8	46.38	7.61

**Table 4 ijerph-19-03686-t004:** Latent growth models for Finnish-born students regarding INSETT intervention.

	Internalizing			Externalizing			Prosocial			Resilience		
	β	SE	*p*	β	SE	*p*	β	SE	*p*	β	SE	*p*
Direct effect												
POST ON intervention	−0.006	0.008	0.428	0.000	0.004	0.934	0.001	0.002	0.651	−0.009	0.013	0.456
PRE ON intervention	0.109	0.381	0.775	0.267	0.156	0.087	−0.181	0.103	0.078	−0.444	0.596	0.457
PRE with POST	0.010	0.019	0.598	0.011	0.019	0.576	−0.011	0.005	0.022	0.022	0.093	0.816
PRE	4.909	0.325	0.000	4.668	0.108	0.000	7.425	0.023	0.000	47.263	0.306	0.000
POST	0.006	0.006	0.300	−0.003	0.003	0.406	0.001	0.001	0.317	0.015	0.008	0.056
*n*	1499			1499			1499			1478		
Log likelihood H0	−7788.937			−7592.931			−6017.435			−9808.834		
Effects with moderators												
POST ON gender	0.008	0.003	0.009	0.004	0.011	0.728	0.002	0.005	0.657	−0.036	0.013	0.006
POST ON gender * intervention	−0.006	0.008	0.446	0.003	0.012	0.769	0.001	0.007	0.848	0.013	0.013	0.317
PRE ON gender	1.969	0.202	0.000	0.225	0.215	0.297	1.055	0.090	0.000	1.802	0.456	0.000
POST ON intervention	−0.003	0.007	0.644	−0.002	0.009	0.833	0.000	0.004	0.974	−0.017	0.017	0.306
PRE ON intervention	0.136	0.327	0.678	0.284	0.155	0.067	−0.160	0.104	0.123	−0.274	0.579	0.636
PRE with POST	0.011	0.018	0.520	0.012	0.019	0.531	−0.015	0.008	0.057	0.033	0.097	0.736
PRE	3.896	0.285	0.000	4.521	0.134	0.000	7.007	0.043	0.000	46.361	0.414	0.000
POST	0.001	0.006	0.830	−0.005	0.008	0.571	0.000	0.002	0.941	0.034	0.014	0.019
*n*	1463			1463			1463			1444		
Log likelihood H0	−7526.121			−7417.897			−5783.924			−9566.129		
POST ON age	0.020	0.002	0.000	0.004	0.003	0.544	0.002	0.001	0.049	−0.014	0.008	0.099
POST ON age * intervention	**−0.014**	**0.003**	**0.000**	−0.002	0.003	0.544	−0.004	0.004	0.334	0.007	0.014	0.597
PRE ON age	0.205	0.127	0.105	0.182	0.130	0.162	−0.133	0.043	0.002	−0.573	0.233	0.014
POST ON intervention	−0.010	0.007	0.167	0.000	0.003	0.967	0.000	0.003	0.861	−0.007	0.013	0.560
PRE ON intervention	0.072	0.357	0.840	0.250	0.162	0.122	−0.142	0.121	0.239	−0.317	0.580	0.585
PRE with POST	0.002	0.018	0.918	0.010	0.020	0.620	−0.011	0.006	0.039	0.022	0.097	0.817
PRE	4.944	0.291	0.000	4.696	0.111	0.000	7.383	0.023	0.000	47.189	0.229	0.000
POST	0.010	0.005	0.032	−0.003	0.001	0.014	0.002	0.001	0.001	0.015	0.005	0.001
*n*	1369			1369			1369			1349		
Log likelihood H0	−7111.075			−6937.750			−5503.487			−8956.760		
POST ON daily stressors (DS)	−0.015	0.003	0.000	−0.014	0.003	0.000	0.010	0.005	0.025	0.030	0.008	0.000
POST ON DS * intervention	**0.016**	**0.007**	**0.021**	0.007	0.007	0.308	−0.008	0.007	0.245	−0.016	0.022	0.469
PRE ON DS	2.150	0.597	0.000	2.397	0.536	0.000	−0.924	0.166	0.000	−7.232	10.265	0.000
POST ON intervention	−0.006	0.008	0.418	−0.001	0.004	0.881	0.001	0.003	0.690	−0.009	0.013	0.483
PRE ON intervention	0.098	0.392	0.804	0.274	0.154	0.076	−0.173	0.124	0.162	−0.450	0.569	0.429
PRE with POST	0.009	0.017	0.622	0.012	0.018	0.514	−0.015	0.007	0.050	0.025	0.080	0.758
PRE	4.925	0.338	0.000	4.667	0.105	0.000	7.527	0.029	0.000	47.294	0.264	0.000
POST	0.006	0.006	0.300	−0.002	0.003	0.447	0.001	0.001	0.190	0.014	0.008	0.072
*n*	1482			1482			1482			1462		
Log likelihood H0	−7681.999			−7467.565			−5917.933			−9595.119		
POST ON discrimination (Disc)	0.010	0.007	0.139	0.007	0.014	0.631	0.006	0.006	0.267	0.053	0.018	0.004
POST ON Disc * intervention	−0.022	0.017	0.176	−0.016	0.016	0.334	**−0.016**	**0.007**	**0.026**	0.000	0.026	0.994
PRE ON Disc	2.332	0.380	0.000	1.889	0.346	0.000	−0.265	0.184	0.150	−5.383	0.610	0.000
POST ON intervention	−0.004	0.007	0.561	0.001	0.005	0.869	0.003	0.002	0.263	−0.007	0.012	0.540
PRE ON intervention	0.162	0.345	0.638	0.312	0.136	0.022	−0.196	0.111	0.078	−0.578	0.589	0.326
PRE with POST	0.008	0.019	0.672	0.011	0.017	0.529	−0.012	0.005	0.028	0.037	0.089	0.683
PRE	4.673	0.263	0.000	4.468	0.067	0.000	7.470	0.049	0.000	47.819	0.232	0.000
POST	0.006	0.006	0.325	−0.003	0.004	0.516	0.000	0.001	0.921	0.008	0.006	0.176
*n*	1471			1471			1471			1455		
Log likelihood H0	−7657.246			−7465.568			−5932.385			−9668.723		

Note: The results are based on latent growth analyses, where POST: Latent slope, PRE: Latent intercept, ON: Regressed on, with: Covariance, *n*: Sample size, variable * variable: Interaction, β: Unstandardized regression coefficient, SE: Standardized error, *p*: *p*-value (<0.050 considered significant and bolded for interactions indicating results from moderation analyses).

**Table 5 ijerph-19-03686-t005:** Latent growth models for immigrant students regarding INSETT intervention.

	Internalizing			Externalizing			Prosocial			Resilience		
	β	SE	*p*	β	SE	*p*	β	SE	*p*	β	SE	*p*
Direct effect												
POST ON intervention	0.006	0.013	0.641	0.029	0.018	0.096	−0.023	0.014	0.106	0.005	0.054	0.931
PRE ON intervention	−0.689	0.762	0.366	−0.799	0.630	0.205	0.238	0.535	0.656	−1.276	1.946	0.512
PRE with POST	−0.048	0.050	0.339	0.006	0.032	0.856	−0.037	0.067	0.579	0.022	0.443	0.960
PRE	6.134	0.704	0.000	5.584	0.592	0.000	7.627	0.379	0.000	46.063	1.548	0.000
POST	0.006	0.008	0.464	−0.015	0.015	0.302	0.010	0.012	0.397	0.019	0.037	0.606
*n*	164			164			164			159		
Log likelihood H0	−809.856			−772.182			−637.981			−982.169		
Effects with moderators												
POST ON gender	0.025	0.014	0.077	0.014	0.023	0.521	−0.007	0.011	0.529	0.002	0.046	0.957
POST ON gender * intervention	0.002	0.015	0.884	0.000	0.025	0.993	0.018	0.011	0.122	0.064	0.053	0.224
PRE ON gender	1.057	0.523	0.043	−0.123	0.425	0.772	0.472	0.353	0.181	1.340	0.711	0.060
POST ON intervention	0.007	0.016	0.670	0.034	0.025	0.174	−0.035	0.017	0.040	−0.012	0.072	0.865
PRE ON intervention	−0.670	0.768	0.383	−0.830	0.574	0.149	0.286	0.530	0.589	−1.466	1.821	0.421
PRE with POST	−0.045	0.050	0.371	0.017	0.029	0.564	−0.035	0.068	0.602	0.115	0.422	0.784
PRE	5.627	0.655	0.000	5.675	0.534	0.000	7.336	0.454	0.000	45.561	1.379	0.000
POST	−0.008	0.013	0.520	−0.026	0.024	0.273	0.018	0.017	0.287	0.009	0.054	0.874
*n*	161			161			161			156		
Log likelihood H0	−789.807			−755.807			−623.669			−964.881		
POST ON age	0.004	0.009	0.668	−0.017	0.021	0.416	−0.007	0.011	0.512	−0.007	0.026	0.792
POST ON age * intervention	0.004	0.013	0.793	0.021	0.021	0.321	0.008	0.011	0.496	0.013	0.030	0.679
PRE ON age	0.049	0.242	0.840	−0.017	0.228	0.940	0.410	0.243	0.092	−0.107	0.784	0.891
POST ON intervention	0.006	0.016	0.697	0.038	0.015	0.013	−0.025	0.012	0.042	−0.003	0.040	0.940
PRE ON intervention	−0.046	0.600	0.939	−0.864	0.566	0.127	0.385	0.549	0.483	−2.241	1.604	0.162
PRE with POST	−0.055	0.065	0.393	−0.007	0.036	0.845	−0.039	0.064	0.544	0.059	0.368	0.872
PRE	5.502	0.521	0.000	5.679	0.519	0.000	7.408	0.387	0.000	46.978	1.241	0.000
POST	0.010	0.011	0.389	−0.023	0.013	0.071	0.012	0.010	0.207	0.022	0.026	0.401
*n*	147			147			147			142		
Log likelihood H0	−724.240			−695.748			−573.998			−872.768		
POST ON DSSYR	−0.002	0.009	0.806	−0.029	0.017	0.087	−0.024	0.013	0.076	0.059	0.024	0.016
POST ON DSSYR * intervention	0.015	0.009	0.087	**0.048**	**0.008**	**0.000**	0.009	0.019	0.639	−0.087	0.051	0.090
PRE ON DSSYR	1.360	0.409	0.001	1.417	0.503	0.005	−0.185	0.360	0.607	−4.833	0.608	0.000
POST ON intervention	0.005	0.012	0.673	0.029	0.017	0.087	−0.022	0.015	0.150	−0.002	0.053	0.974
PRE ON intervention	−0.858	0.737	0.244	−0.845	0.607	0.163	0.235	0.546	0.667	−1.073	1.913	0.575
PRE with POST	−0.052	0.053	0.326	−0.008	0.028	0.778	−0.054	0.134	0.687	0.035	0.417	0.933
PRE	6.315	0.671	0.000	5.652	0.567	0.000	7.629	0.387	0.000	45.902	1.528	0.000
POST	0.007	0.009	0.431	−0.015	0.014	0.292	0.008	0.013	0.509	0.022	0.038	0.555
*n*	160			160			160			156		
Log likelihood H0	−791.074			−753.518			−626.876			−955.315		
POST ON Disc	−0.012	0.014	0.383	−0.015	0.030	0.618	0.017	0.025	0.498	0.003	0.043	0.950
POST ON Disc * intervention	0.007	0.016	0.651	0.039	0.029	0.175	−0.031	0.025	0.208	0.028	0.032	0.382
PRE ON Disc	2.531	0.445	0.000	0.844	0.540	0.118	0.369	0.357	0.302	−2.060	1.752	0.240
POST ON intervention	−0.004	0.017	0.827	0.012	0.025	0.643	−0.010	0.016	0.538	−0.009	0.062	0.882
PRE ON intervention	−0.766	0.643	0.234	−0.914	0.655	0.163	0.220	0.544	0.686	−1.236	2.075	0.552
PRE with POST	−0.047	0.047	0.321	0.007	0.029	0.793	−0.045	0.079	0.567	0.018	0.428	0.966
PRE	5.157	0.690	0.000	5.353	0.666	0.000	7.513	0.363	0.000	46.937	1.940	0.000
POST	0.014	0.015	0.350	−0.010	0.025	0.688	0.003	0.015	0.820	0.020	0.043	0.642
*n*	156			156			156			152		
Log likelihood H0	−757.329			−723.425			−598.039			−946.160		

Note: The results are based on latent growth analyses, where POST: Latent slope, PRE: Latent intercept, ON: Regressed on, with: Covariance, *n*: Sample size, variable * variable: Interaction, β: Unstandardized regression coefficient, SE: Standardized error, *p*: *p*-value (<0.050 considered significant and bolded for interactions indicating results from moderation analyses).

**Table 6 ijerph-19-03686-t006:** Latent growth models for Finnish-born and immigrant students regarding PIER intervention.

	Internalizing			Externalizing			Prosocial			Resilience		
	β	SE	*p*	β	SE	*p*	β	SE	*p*	β	SE	*p*
Direct effect												
POST ON intervention	0.001	0.013	0.911	0.008	0.009	0.348	0.000	0.002	0.828	0.013	0.031	0.674
PRE ON intervention	−1.264	0.698	0.070	−0.498	0.537	0.354	−0.066	0.396	0.868	0.333	0.893	0.709
PRE with POST	0.025	0.029	0.379	0.016	0.023	0.492	−0.003	0.006	0.628	−0.155	0.599	0.795
PRE	5.360	0.365	0.000	5.191	0.256	0.000	7.487	0.051	0.000	47.147	0.524	0.000
POST	0.007	0.007	0.287	0.001	0.005	0.822	0.001	0.003	0.633	−0.008	0.021	0.703
*n*	297			297			297			294		
Log likelihood H0	−1766.211			−1713.106			−1353.047			−2162.635		
Effects with moderators												
POST ON gender	−0.003	0.012	0.794	0.002	0.004	0.637	−0.005	0.009	0.557	0.011	0.042	0.787
POST ON gender * intervention	0.010	0.033	0.767	**0.010**	**0.004**	**0.008**	0.002	0.008	0.799	**−0.084**	**0.037**	**0.023**
PRE ON gender	2.427	0.215	0.000	0.819	0.482	0.089	0.807	0.046	0.000	0.810	0.304	0.008
POST ON intervention	−0.002	0.001	0.074	0.005	0.008	0.548	−0.003	0.005	0.551	0.045	0.044	0.311
PRE ON intervention	−0.917	0.648	0.157	−0.410	0.442	0.354	0.080	0.420	0.849	0.347	0.938	0.712
PRE with POST	0.013	0.027	0.621	0.013	0.028	0.655	−0.001	0.004	0.863	−0.172	0.572	0.763
PRE	3.863	0.270	0.000	4.650	0.336	0.000	7.008	0.079	0.000	46.715	0.529	0.000
POST	0.009	0.008	0.235	0.000	0.006	0.998	0.005	0.009	0.570	−0.013	0.046	0.780
*n*	288			288			288			285		
Log likelihood H0	−1695.440			−1651.720			−1306.041			−2099.013		
POST ON age	0.011	0.004	0.006	−0.001	0.003	0.834	0.002	0.003	0.503	0.016	0.020	0.437
POST ON age * intervention	**−0.018**	**0.002**	**0.000**	**−0.015**	**0.008**	**0.043**	0.001	0.003	0.681	**0.025**	**0.005**	**0.000**
PRE ON age	0.267	0.217	0.218	0.783	0.016	0.000	−0.253	0.076	0.001	−2.027	0.212	0.000
POST ON intervention	−0.004	0.014	0.774	0.003	0.011	0.773	0.003	0.004	0.402	0.025	0.033	0.456
PRE ON intervention	−0.941	0.797	0.238	−0.043	0.605	0.944	−0.239	0.488	0.625	−0.687	1.474	0.641
PRE with POST	0.017	0.031	0.576	0.016	0.016	0.306	−0.003	0.007	0.643	−0.196	0.722	0.786
PRE	5.293	0.402	0.000	5.148	0.245	0.000	7.525	0.110	0.000	47.525	0.899	0.0000
POST	0.011	0.004	0.006	0.001	0.006	0.827	0.001	0.004	0.702	−0.012	0.029	0.675
*n*	263			263			263			260		
Loglikelihood H0	−1579.770			−1520.416			−1203.159			−1919.866		
POST ON daily stressors (DS)	−0.015	0.016	00.349	0.018	0.010	0.068	−0.024	0.009	0.009	−0.011	0.054	0.831
POST ON DS * intervention	−0.013	0.013	00.298	−0.012	0.010	0.232	0.005	0.008	0.510	−0.004	0.052	0.934
PRE ON DS	3.584	1.368	0.009	3.096	0.315	0.000	−0.326	0.242	0.177	−5.948	3.276	0.069
POST ON intervention	0.003	0.011	0.795	0.008	0.009	0.352	0.001	0.002	0.524	0.016	0.030	0.601
PRE ON intervention	−1.372	0.393	0.000	−0.589	0.221	0.008	−0.096	0.380	0.801	0.334	0.689	0.627
PRE with POST	0.022	0.035	0.531	0.006	0.032	0.861	−0.005	0.007	0.526	−0.182	0.464	0.695
PRE	5.377	0.233	0.000	5.234	0.200	0.000	7.521	0.052	0.000	47.395	0.225	0.000
POST	0.007	0.008	0.359	0.001	0.006	0.806	0.000	0.003	0.926	−0.016	0.020	0.419
*n*	291			291			291			288		
Log likelihood H0	−1718.125			−1662.761			−1324.076			−2089.646		
POST ON discrimination (Disc)	−0.008	0.011	0.467	−0.012	0.006	0.048	−0.016	0.008	0.050	0.013	0.026	0.632
POST ON Disc * intervention	−0.010	0.011	0.338	0.015	0.007	0.021	0.027	0.008	0.001	0.015	0.020	0.467
PRE ON Disc	1.840	0.188	0.000	1.859	0.644	0.004	−0.086	0.126	0.493	−3.187	1.739	0.067
POST ON intervention	0.000	0.015	0.989	0.006	0.009	0.541	−0.004	0.003	0.252	0.019	0.033	0.567
PRE ON intervention	−1.212	0.608	0.046	−0.488	0.395	0.217	−0.109	0.306	0.723	0.190	0.582	0.744
PRE with POST	0.012	0.028	0.660	0.017	0.016	0.278	−0.002	0.006	0.776	−0.134	0.505	0.790
PRE	5.059	0.368	0.000	4.877	0.340	0.000	7.522	0.043	0.000	47.916	0.434	0.000
POST	0.011	0.005	0.018	0.005	0.005	0.314	0.002	0.003	0.419	−0.020	0.015	0.185
*n*	272			272			272			270		
Log likelihood H0	−1635.438			−1579.253			−1251.768			−1994.437		
POST ON immigrant status	0.004	0.021	0.858	−0.023	0.023	0.304	−0.001	0.015	0.956	0.036	0.052	0.493
POST ON immigr * intervention	0.019	0.016	0.242	0.011	0.037	0.771	0.019	0.018	0.297	0.022	0.016	0.187
PRE ON immigrant status	0.767	0.289	0.008	1.166	0.856	0.173	−0.483	0.171	0.012	−2.390	2.088	0.252
POST ON intervention	−0.002	0.012	0.893	0.007	0.011	0.510	−0.002	0.003	0.347	0.012	0.030	0.693
PRE ON intervention	−1.208	0.672	0.072	−0.513	0.480	0.285	0.001	0.326	0.998	0.133	0.950	0.889
PRE with POST	0.009	0.027	0.737	0.020	0.026	0.436	−0.002	0.006	0.704	−0.131	0.542	0.809
PRE	5.214	0.431	0.000	5.064	0.325	0.000	7.544	0.063	0.000	47.483	0.694	0.000
POST	0.008	0.007	0.199	0.003	0.003	0.319	0.001	0.003	0.777	−0.011	0.017	0.537
*n*	289			289			289			286		
Log likelihood H0	−1720.189			−1672.041			−1310.678			−2108.512		

Note: The results are based on latent growth analyses, where POST: Latent slope, PRE: Latent intercept, ON: Regressed on, with: Covariance, *n*: Sample size, variable * variable: Interaction, β: Unstandardized regression coefficient, SE: Standardized error, *p*: *p*-value (<0.050 considered significant and bolded).

## Data Availability

We have not yet achieved the data (as the international project is still in progress). At the moment, the data that support the findings are available on request, but are not public due to privacy and ethical restrictions.

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
