# Peer review of "Effectiveness of Promotive and Preventive Psychosocial Interventions on Improving the Mental Health of Finnish-Born and Immigrant Adolescents"

_ijerph, 2022, doi:10.3390/ijerph19063686_

Round 1

Reviewer 1 Report

I am immediately struck at the start by the breadth of this research that stretches out through many countries  (Finland, Denmark, Great Britain, and Belgium) in terms of contributing authors/co-authors, and five universities (Turku, Tampere, Sussex, Copenhagen, and Ghent). That takes an effective organizing of research, individuals, and distribution of labor, and moving them to a wholesome focus. The research was successful in this because the research is coherent. Though the main focus is on enhancing the mental health, emotional well-being, and the social skills of mostly refugee and immigrant students of adolescent age in Finnish secondary schools, native students of the same age-group are included for the purpose of comparison and to validate conclusions reached. It is a good approach since the trauma of war and the toll migration takes on these students would be different from the more sedate and relatively tranquil life of native Finnish students. But both categories of students do experience some cultural shock--the shock of the other and of readjustment to a new reality. It is a good topic to focus on and it is timely, too. The main force that limits the strength, not the effectiveness, of the research is one outside the authors' control--COVID-19 and its Greek-alphabet variants. One can only wonder if the results would be different if the schools had remained in-person for evidence-and-response gathering instead of the online isolation of instructions that made physical contacts impossible. One also wonders if the virus itself has heightened the post-traumatic stress disorder and other emotional problems experienced by refugee students and their families. What psychological damage would be borne by immigrant parents and children should a family member die of COVID after the long journey to escape from death and carnage that had ravaged their nations of origin? It seems to me that the authors may want to consider a post-COVID assessment of the mental and cultural health of immigrant students and their families compared to native Finnish counterparts as a segue to their current research. This is not meant as a condition for accepting this present research which, on its own, is robust and interventionist and whose end could be therapeutic when fully realized. The research hits the target in establishing the necessity of teachers and peer groups to be trained to intervene successfully in improving the mental health of Finnish citizens and immigrant residents. This is an important research contribution to the social and psychological well-being of Finns and their immigrant neighbors as both sides learn to adjust well to their new reality.

The review of existing studies related to this research is very good. It was nicely done by the authors to create an interventionist space to show the importance and relevance of their current research. I do have a question for the authors here. If "Ten of the 14 studies in the review were conducted at schools," where were the remaining four studies conducted? This should be accounted for to give readers the full information. The authors also used the phrase, "teacher self-efficacy," repeatedly without explaining it. Though teachers and educators may readily know this phrase as a shared professional language, the meaning may not be so obvious to government agencies and workers in other professions. It should be explained the first time it is used, at least in a footnote if it may disrupt the body of the essay proper.

The research methods are clearly explained. They follow acceptable standards that would yield acceptable and convincing results. The authors should shed more light on why only "19% [of] students with immigrant backgrounds" were invited to participate in the study. Since they were somehow important to the success of the study, a stronger effort should have been made to include more of them at the beginning of the research. That would have helped, perhaps, to take care of the attrition rate expected in a research of this nature.  The results reached are, however, sound and in no way compromised. They are supported by the evidence presented. The overall method is also ethically sound.

The authors are direct in revealing the limits of their study--that "the initial research aims of searching the mediating mechanisms and analyzing all three levels of interventions were not fulfilled." I have already addressed this above by pointing out that the percentage of immigrant students who participated in the study should have been more. The authors should take this into consideration in their future research. COVID-19 placed a limit on teachers' participation, especially their response rate. This is understandable because of the pressure on teachers to develop and deliver online and Zoom classes for their students. This was a time at which technology was stretched to the limits and people's mental health stretched perhaps as never before at the same time in the whole world.

Authors should check page 7 (under "3.4. Intervention dosage") and page 17 (under "Institutional Review Board Statement") for some minor editing.

Overall, it is a good research worthy of publication.

Author Response

I am immediately struck at the start by the breadth of this research that stretches out through many countries  (Finland, Denmark, Great Britain, and Belgium) in terms of contributing authors/co-authors, and five universities (Turku, Tampere, Sussex, Copenhagen, and Ghent). That takes an effective organizing of research, individuals, and distribution of labor, and moving them to a wholesome focus. The research was successful in this because the research is coherent. Though the main focus is on enhancing the mental health, emotional well-being, and the social skills of mostly refugee and immigrant students of adolescent age in Finnish secondary schools, native students of the same age-group are included for the purpose of comparison and to validate conclusions reached. It is a good approach since the trauma of war and the toll migration takes on these students would be different from the more sedate and relatively tranquil life of native Finnish students. But both categories of students do experience some cultural shock--the shock of the other and of readjustment to a new reality. It is a good topic to focus on and it is timely, too.

Thank you for your supportive comments and insightful suggestions. We are happy that you found the approach relevant. We too feel that having the perspective both native and immigrant students is important. We have tried to address all the issues you raised. The suggestions and questions were very helpful and we feel that the manuscript really got better when implementing these corrections. Thank you once again. We have summarized the corrections we made below:

The main force that limits the strength, not the effectiveness, of the research is one outside the authors' control--COVID-19 and its Greek-alphabet variants. One can only wonder if the results would be different if the schools had remained in-person for evidence-and-response gathering instead of the online isolation of instructions that made physical contacts impossible. One also wonders if the virus itself has heightened the post-traumatic stress disorder and other emotional problems experienced by refugee students and their families. What psychological damage would be borne by immigrant parents and children should a family member die of COVID after the long journey to escape from death and carnage that had ravaged their nations of origin?

You are absolutely right that the strength of our research was challenged by the COVID. The issues you raised are exactly those we have wondered too. We felt unable to speculate what have happened in those cases (research-wise and also related to the individual students and their families.  However we have tried to be transparent with the effects of covid on our study in three ways: 1) At page 9 we state that “Due to the Covid-19 school closures, the T2 and T3 data collection were guided online.” 2) At page 21 we state that “the onset of the Covid-19 pandemic during the post-intervention and follow-up assessments posed additional challenges for these endeavors.” 3) According to your comments we have now also added one more statement in the discussion text at page 21 saying that : “ In addition to affecting the response rates of especially the teachers, we can only speculate on the other effects Covid-19 might have had on our results.”

It seems to me that the authors may want to consider a post-COVID assessment of the mental and cultural health of immigrant students and their families compared to native Finnish counterparts as a segue to their current research. This is not meant as a condition for accepting this present research which, on its own, is robust and interventionist and whose end could be therapeutic when fully realized. The research hits the target in establishing the necessity of teachers and peer groups to be trained to intervene successfully in improving the mental health of Finnish citizens and immigrant residents. This is an important research contribution to the social and psychological well-being of Finns and their immigrant neighbors as both sides learn to adjust well to their new reality.

Thank you for your words. We really feel that the best thing our study can offer is the knowledge about the wellbeing of adolescents in this situation, and the insights of interventions taking place at school

The review of existing studies related to this research is very good. It was nicely done by the authors to create an interventionist space to show the importance and relevance of their current research. I do have a question for the authors here. If "Ten of the 14 studies in the review were conducted at schools," where were the remaining four studies conducted? This should be accounted for to give readers the full information. The authors also used the phrase, "teacher self-efficacy," repeatedly without explaining it. Though teachers and educators may readily know this phrase as a shared professional language, the meaning may not be so obvious to government agencies and workers in other professions. It should be explained the first time it is used, at least in a footnote if it may disrupt the body of the essay proper.

Thank you for the close reading of our manuscript. We apologize for the mistake in this matter. Indeed,14/20 studies were conducted at schools, 1/20 college ja 5/20 community settings. We have now added the information about the other study settings included in the review on p. 7 by saying that “Fifteen out of 20 studies in the review were conducted at schools or a college, and five in community settings, which may indicate the importance of supportive peer work especially in a classroom context”. You are also right that we should have been explained the content of teacher self-efficacy concept. This has now been added at page 6 in the following way: Teacher self-efficacy, i.e. the teachers’ own judgement of their capability in inducing desirable student outcomes, has also been found to improve student academic performance (38) and motivation through instructional support (39)

The research methods are clearly explained. They follow acceptable standards that would yield acceptable and convincing results. The authors should shed more light on why only "19% [of] students with immigrant backgrounds" were invited to participate in the study. Since they were somehow important to the success of the study, a stronger effort should have been made to include more of them at the beginning of the research. That would have helped, perhaps, to take care of the attrition rate expected in a research of this nature.  The results reached are, however, sound and in no way compromised. They are supported by the evidence presented. The overall method is also ethically sound.

We apologize for not being clear about the protocol of inviting the students to the study. It is important to note that we invited all children in schools to participate. 19% of all children in schools had immigrant background. But you are right that more native than immigrant children participated and if we would have had more resources, a stronger effort would have been made to include more of them at the beginning of the research. Unfortunately, our resources were not sufficient to go to the schools even more and boost the recruitment of immigrant students. We have now corrected the text in the following at page 10 way to be more precise about the recruitment process. “In recruitment, school authorities estimated the number of students and the proportion of foreign-born[1] students in all participating classes. We invited all children in schools to participate. The actual number of students that were invited to participate in the study was 3062, with 19% being students with immigrant backgrounds. Yet, 1974 students, of whom 223 were foreign- and 1770 Finnish-born secondary school students (aged 12─17 years) answered to the baseline (T1) questionnaire. While over two-thirds (71%) of the invited Finnish-born students participated, refusal among immigrant students was considerable; only four out of ten (39 %) participated.”

The authors are direct in revealing the limits of their study--that "the initial research aims of searching the mediating mechanisms and analyzing all three levels of interventions were not fulfilled." I have already addressed this above by pointing out that the percentage of immigrant students who participated in the study should have been more. The authors should take this into consideration in their future research. COVID-19 placed a limit on teachers' participation, especially their response rate. This is understandable because of the pressure on teachers to develop and deliver online and Zoom classes for their students. This was a time at which technology was stretched to the limits and people's mental health stretched perhaps as never before at the same time in the whole world.

You are right. We have added now more discussion about this issue at page 22-23 by saying that “Also, the percentage of immigrant students who participated in the study should have been more in order to make strong implications of the differences between the students with different backgrounds. Having more resources to ensure the participation of immigrant students should be taken into consideration in future research.”

Authors should check page 7 (under "3.4. Intervention dosage") and page 17 (under "Institutional Review Board Statement") for some minor editing.

Thank you for the close reading. We have now edited the text and corrected the errors.

Overall, it is a good research worthy of publication.

Thank you, we hope that the corrections we have made, based on your suggestions, improved the paper

Reviewer 2 Report

The article Effectiveness of promotive and preventive psychosocial interventions on improving the mental health of Finnish-born and immigrant adolescents, presents a structured paper, with good scientific writing and clearly presented.

It relies on relevant work of the same scope in its rationale. This means that the bibliography is relevant.

The theme is also current and relevant.

Ethical issues are ensured.

It is based on a very extensive work, where questionnaires that have already been validated were used. Naturally it generated a large amount of data that, in my opinion, was well interpreted.

It was unfortunate that the study had some limitations, such as less involvement from teachers and less participation from immigrant students, which is partially reflected in the results. These occurrences are always part of learning in research processes.

The work could be improved in the conclusion. The conclusion is very short and focuses essentially on the constraints of the work. It should focus mainly on the most relevant results of the research, although in a synthetic way. 

Author Response

The article Effectiveness of promotive and preventive psychosocial interventions on improving the mental health of Finnish-born and immigrant adolescents, presents a structured paper, with good scientific writing and clearly presented.

It relies on relevant work of the same scope in its rationale. This means that the bibliography is relevant.

The theme is also current and relevant.

Ethical issues are ensured.

It is based on a very extensive work, where questionnaires that have already been validated were used. Naturally it generated a large amount of data that, in my opinion, was well interpreted.

Thank you for your kind words and seeing the strength of our paper in the field. Thank you also recognizing the large amount of field work behind the data.

It was unfortunate that the study had some limitations, such as less involvement from teachers and less participation from immigrant students, which is partially reflected in the results. These occurrences are always part of learning in research processes.

Yes, we really feel so too. I think we have learned a lot for the future trials on how to encourage the teachers and subgroups of students to participate.

The work could be improved in the conclusion. The conclusion is very short and focuses essentially on the constraints of the work. It should focus mainly on the most relevant results of the research, although in a synthetic way. 

We agree that the conclusion was too much focused on the limitations. We have now added the following text at the beginning of the Conclusion section to highlight the main findings and practical implications of our study: “The positive effects of interventions in this study were dependent on students’ characteristics such as gender, age, exposure to daily stressors and their discrimination experiences. This means that tailored help is needed in order to support adolescents’ wellbeing at schools. Future studies should continue to search for critical factors that affect the suitability of the interventions for adolescents with different backgrounds.”   

Reviewer 3 Report

My only suggestion is that the authors need to discuss why the low participation rate.  Did they attempt to reach out to students/teachers to more fully understand the participation rates.    On page 8, it reads: "for follow up, they were.....  (I assume this meant for T3 but it is not clear.)

An excellent study, good analysis and the statistical procedures are appropriate.

I found the paper to be well executed, properly documented, presented a good review of the literature and the statistical analysis is properly carried out. It might not have been the statistics I would have used, but I didn't write the paper and do the statistics. The authors faithfully carried out a good research project. I gave you my only two comments and I don't understand why I would criticize the authors when they did an excellent job. All research projects have deficiencies and some are problematic or potentially biased. In the present case, the authors have explained various methodological and statistical choices. Yes, there are cells with very few cases but that is the result of the lack of participants--and that is why I asked that the authors need to address that question.

When scholars carry out good research (never perfect), they should not be criticized for the results they obtain. If the results were not an artifact of the methodology or statistical analysis, then the paper should be published. It has a very important message for policy makers and policy implementation that should be addressed and recognized.

Author Response

My only suggestion is that the authors need to discuss why the low participation rate.  Did they attempt to reach out to students/teachers to more fully understand the participation rates.   

We apologize for not being clear about the protocol of inviting the students to the study and the reasons for the low participating rate. It is important to note that we invited all children in schools to participate. 19% of all children in schools had immigrant background. More native than immigrant children participated and if we would have had more resources, a stronger effort would have been made to include more of them at the beginning of the research. Unfortunately, our resources were not sufficient to go to the schools even more and boost the recruitment of immigrant students. We have now corrected the text in the following at page 10 way to be more precise about the recruitment process. “In recruitment, school authorities estimated the number of students and the proportion of foreign-born[1] students in all participating classes. We invited all children in schools to participate. The actual number of students that were invited to participate in the study was 3062, with 19% being students with immigrant backgrounds. Yet, 1974 students, of whom 223 were foreign- and 1770 Finnish-born secondary school students (aged 12─17 years) answered to the baseline (T1) questionnaire. While over two-thirds (71%) of the invited Finnish-born students participated, refusal among immigrant students was considerable; only four out of ten (39 %) participated.” We also added the following text in the Discussion section: “Also, the percentage of immigrant students who participated in the study should have been more in order to make strong implications of the differences between the students with different backgrounds. Having more resources to ensure the participation of immigrant students should be taken into consideration in future research. In addition to affecting the response rates of especially the teachers, we can only speculate on the other effects Covid-19 might have had on our results.”

On page 8, it reads: "for follow up, they were.....  (I assume this meant for T3 but it is not clear.)

Thank you for the close reading. This sentence at page 8 refers indeed to T3. We have now added the information in brackets.

An excellent study, good analysis and the statistical procedures are appropriate.

I found the paper to be well executed, properly documented, presented a good review of the literature and the statistical analysis is properly carried out. It might not have been the statistics I would have used, but I didn't write the paper and do the statistics. The authors faithfully carried out a good research project. I gave you my only two comments and I don't understand why I would criticize the authors when they did an excellent job. All research projects have deficiencies and some are problematic or potentially biased. In the present case, the authors have explained various methodological and statistical choices. Yes, there are cells with very few cases but that is the result of the lack of participants--and that is why I asked that the authors need to address that question.

When scholars carry out good research (never perfect), they should not be criticized for the results they obtain. If the results were not an artifact of the methodology or statistical analysis, then the paper should be published. It has a very important message for policy makers and policy implementation that should be addressed and recognized.

We highly appreciate your overall statement of our paper. It is indeed true that carrying out a multisite intervention study is seldom perfect and we are happy that you see the value of our results through the limitations we have articulated ourselves too.  We are working with multiple actions to reach out the policy makers and officials working in the field and try to spread the message of these issues

Reviewer 4 Report

Thank you for submitting the manuscript. The introduction provides an adequate overview, while the author(s) failed to show the aim of the study which requires a clearly adequate justification on the last part of introduction.

The literature was not clearly stated mainly on defining the keyword of each topic. The most important part is how the current study would contribute into the existing knowledge literature.

So, formulation on convincing the current study towards the previous research should be made properly. This is important to ensure the reader(s) would see the main contribution to the current literature.

The methodology applied needs some clarification.

Why has this method of data collection been chosen?

How was the sample put together?

How was data analyzed?

With regards to the results and discussion, the results of the study are presented clearly. However, it is not stated how and why the aspects presented have been chosen.

Good luck for the authors. 

I had a good time reading your research paper. The idea is interesting and would encourage this to be bought to light. However, this paper would need more editing and reviewing. Do not lose hope. You are almost there!

Author Response

Thank you for submitting the manuscript. The introduction provides an adequate overview, while the author(s) failed to show the aim of the study which requires a clearly adequate justification on the last part of introduction…. So, formulation on convincing the current study towards the previous research should be made properly. This is important to ensure the reader(s) would see the main contribution to the current literature.

Thank you for your valuable feedback. We agree that the rationale of the study could have been better highlighted at the end of the introduction section. We have now added the following paragraph at page 7: “In sum, there is lack of research studying the effects of school-based interventions, focusing either on teacher competence or peer relations, on adolescent’s wellbeing. This is especially true concerning the samples with native and immigrant students. In this study, we aim to find out whether interventions that aim to improve teacher’s competence and self-efficacy or peer relations among students in diverse classrooms are effective in supporting students’ mental health.”

The literature was not clearly stated mainly on defining the keyword of each topic. The most important part is how the current study would contribute into the existing knowledge literature.

Thank you for pointing out these shortcomings in the introduction text. We have now tried to define the key concepts in more detail. First, at page 4 immigrant young people is defined as foreign-born adolescent living in the new country of residence. Second, at page 4 we spell out the acronym PTSD as post-traumatic stress disorder. Third, at page 6 Teacher self-efficacy is defined as teachers’ own judgement of their capability in inducing desirable student outcomes. See also the response above regarding the aim of the study and the contribution to the existing literature. Thus, we aimed to articulate more clearly the gap in the existing literature and our study aims in relation to that gap (at page 7).

The methodology applied needs some clarification. How was the sample put together? Why has this method of data collection been chosen? How was data analyzed?

We have tried to describe the analysis in detail at page 15. We used the growth curve model (GCM) with individually varying timepoints of observation (so-called unstructured GCM). The unstructured GCM has produced less biased results in several simulation studies, which matched our data with varying timepoints of observation. The unstructured GCM model is a multi-level model and also allows to take clustering into account.  In this revised version of the manuscript we explain the data collection and data management in more detail at page 15. We have added the following text: “Data was collected in Lime Survey in a long form, i.e., each participant's response was unique. Participants received an identity number which then was used to combine responses from different assessment points. Identity numbers in data file were checked manually for errors so that the correct responses from the same respondent were brought together. Each time point was then merged one by one so that we ended up with a wide format data of three time points. “ We hope that these clarifications are helpful and that the methodological section is improved.

With regards to the results and discussion, the results of the study are presented clearly. However, it is not stated how and why the aspects presented have been chosen.

We have constructed the Discussion text in the following way: First, we introduce the main findings of our intervention trial. Second, we have highlighted how these results were dependent on adolescent characteristics such as age, gender, exposure to daily stressors in their lives and experiences of discrimination. Third, we reflect the shortcomings and gaps related to our design, data collection and other issues. In other words, we have tried to include those results in the discussion section that were the core findings in answering our research questions. In this revised version we have also broadened the conclusion section and added the most important findings.  

Good luck for the authors. 

I had a good time reading your research paper. The idea is interesting and would encourage this to be bought to light. However, this paper would need more editing and reviewing. Do not lose hope. You are almost there!

Thank you so much for these encouraging words. We have tried to re-write the manuscript according to your and other reviewers’ comments and hope that it is now improved and ready to be published
